# A Possibility to Use Selected Crop Post-Extraction Wastes to Improve the Composition of Cultivated Mushroom *Pleurotus citrinopileatus*

**DOI:** 10.3390/jof7110894

**Published:** 2021-10-22

**Authors:** Zuzanna Magdziak, Monika Gąsecka, Kinga Stuper-Szablewska, Marek Siwulski, Sylwia Budzyńska, Agnieszka Jasińska, Przemysław Niedzielski, Pavel Kalač, Mirosław Mleczek

**Affiliations:** 1Department of Chemistry, Poznan University of Life Sciences, 60-625 Poznań, Poland; monika.gasecka@up.poznan.pl (M.G.); kinga.stuper@up.poznan.pl (K.S.-S.); sylwia.budzynska@up.poznan.pl (S.B.); miroslaw.mleczek@up.poznan.pl (M.M.); 2Department of Vegetable Crops, Poznan University of Life Sciences, 60-025 Poznań, Poland; marek.siwulski@up.poznan.pl (M.S.); agnieszka.jasinska@up.poznan.pl (A.J.); 3Faculty of Chemistry, Adam Mickiewicz University in Poznań, 61-614 Poznań, Poland; pnied@amu.edu.pl; 4Department of Applied Chemistry, Faculty of Agriculture, University of South Bohemia, 370 05 České Budějovice, Czech Republic; kalac@zf.jcu.cz

**Keywords:** *Pleurotus citrinopileatus*, trace elements, low-molecular-weight organic acids, phenolic compounds, antioxidative effect, organic wastes

## Abstract

A cultivated mushroom species, *Pleurotus citrinopileatus*, is becoming increasingly popular thanks to its attractive colour and medicinal properties. In this study, *P. citrinopileatus* was grown in a cultivation medium enriched with wheat bran (WB), thymus post-extraction waste (TPEW) and pumpkin post-extraction waste (PPEW) products. The study showed that the post-extraction wastes are a crucial factor determining the accumulation of minerals, the content/profile of low-molecular-weight organic acids (LMWOAs) and phenolic compounds in fruit bodies, thereby increasing their nutritional value. The use of the waste materials significantly increased LMWOAs contents. The sum of LMWOAs under all cultivation mediums increased, especially quinic, malic and citric acids under the 20% PPEW, 25 and 50% TPEW addition. Total phenolic content, phenolic content, as well as the composition and scavenging effect on DPPH radicals, were strongly dependent on the used substrate. The control variant was poor in phenolic compounds, while the supplementation increased the contents and diversity of these metabolites. In the control, only four phenolic compounds were quantified (chlorogenic, gallic, syringic and vanillic acids), while in the supplemented substrates up to 14 different phenolic compounds (caffeic, chlorogenic, *p*-coumaric, 2,5-dihydroxybenzoic acid, ferulic, gallic, protocatechuic, salicylic, sinapic, syringic, *trans*-cinnamic and vanillic acids, catechin and rutin).

## 1. Introduction

*Pleurotus citrinopileatus* or *P. cornucopiae* var. *citrinopileatus*, also known by its common name Golden oyster mushroom, is an edible mushroom belonging to the genus *Pleurotus*, a member of the Pleurotaceae family and it is commonly found in Asia [1]. Together with *P. ostreatus* it comprised 25% of Chinese mushroom production in 2010. According to FAOSTAT (2016) [2] and USDA [3], world production of the *Pleurotus* genus reached 27%, which makes this mushroom the second most commonly cultivated mushroom species.

*Pleurotus citrinopileatus* is cultivated on a commercial scale mainly in India, China, Korea and Japan. Due to its high thermal requirements, *P. citrinopileatus* is grown in tropical or subtropical areas, or even in temperate regions during the summer season. The substrates used for cultivation depend on locally available agricultural wastes such as sugar cane pomace, sawdust, rice straw, cottonseed cake, nut hulls, coconut fiber waste and date palm [4,5,6,7,8,9,10,11], but most commonly it is grown on a substrate containing straw, sawdust or ground corn cobs [4,5,8,12]. The golden oyster mushroom exhibits a high level of polymer degradation which makes it an easy-to-grow species, while its mycelium is characterized by rapid growth [12,13,14,15]. In Poland wheat or rye straw sawdust of deciduous trees enriched with cereal bran and ground corncobs are mainly used in the cultivation of *P. citrinopileatus* [5,16]. *Pleurotus citrinopileatus* may be an alternative for cultivation to the most commonly grown *Pleurotus ostreatus*, *Pleurotus pulmonarius* and *Pleurotus precoce* thanks to its attractive colour, its use as an excellent ingredient of mushroom mixtures and high tolerance to extreme cultivation temperatures [5].

*Pleurotus* species have an ability to accumulate major essential elements (e.g., Ca, K or Mg) and essential trace elements (e.g., B, Cu, Cr or Mo) from the different chemical characteristics of substrates and thus they may present a dietary supplement [17]. *Pleurotus citrinopileatus* is a source of essential elements and bioactive compounds with medicinal properties. Recent studies indicate considerable contents of such elements as Al, Ca, K, Mg, Na, P, Cl, Cu, Fe, Mn, Rb, Re, S, Si, Ti, U and Zn in *P. citrinopileatus* fruit bodies [18,19,20] and the highest antioxidant activity among the tested species, i.e., *P. ostreatus*, *P. pulmonarius* and their strains [21].

The main active compounds identified in *P. citrinopileatus* fruit bodies include triacylglycerols, terpenes, steroids, glucans, polysaccharides, proteins and their derivatives, amines, but also flavonoids and phenols [22,23,24,25,26,27]. Their anticancer, antihyperglycaemic, immunostimulatory, antioxidant and anti-inflammatory properties were demonstrated by many authors [4,8,22,28,29,30,31,32,33,34,35,36,37,38,39]. Many studies focused on the antioxidant properties of phenolics contained in mushrooms of the *Pleurotus* genus [21,23,40,41,42]. A high correlation between antioxidant capacity, phenolic and flavonoid contents indicates that phenolic compounds may be the main contributors to the antioxidant activities of edible mushrooms [43,44,45]. This study reflects the recent studies of Peraza et al. [21], where *P. citrinopileatus* showed the highest antioxidant activity among the investigated species, *P. ostreatus*, *P. pulmonarius* and their strains. This study also highlights the importance of cultivation substrates and the activity of total flavonoid contents and antioxidants of cultivated mushrooms in relation to the use of wild grass species rather than commercial wheat straw [45].

Studies conducted in the last years have shown that the yielding of *P. citrinopileatus* can be increased by enrichment of straw or sawdust substrate with organic additives, which are the source of nitrogen [7,16,46,47,48,49]. In addition, other types of additives may be beneficial for mushroom growth and yielding. Siwulski et al. [50] proved that the addition of *Gingko biloba* extract to beech sawdust substrate stimulates the growth of mycelium. However, any modification may significantly affect the composition of the produced fruit bodies. Depending on the substrate and its additives, the ability to absorb major and trace elements, the content of phenolic compounds, as well as low-molecular-weight organic acids (LMWOAs) may significantly change in mushrooms. To our knowledge, there is no information concerning the profile and content of LMWOAs in *P. citrinopileatus*. The LMWOAs in mushroom fruit bodies play a decisive role in determining organoleptic properties such as taste and smell [51,52]. They change slowly during storage and may have a protective role against various diseases [53] and chelate metals [54].

There are few reports indicating changes in the contents of nutritional and trace elements, as well as the content and composition of phenolics and LMWOAs in *P. citrinopileatus* after enrichment of mycelium using different waste products. For this reason, the aim of this study is to determine the chemical characteristics of *P. citrinopileatus* fruit bodies (multi-elemental analysis, total phenolic content, phenolic compounds and LMWOAs).

## 2. Materials and Methods

### 2.1. Experimental Material and Experimental Design

In the experiment, *P. citrinopileatus* strain B83 from the Mushrooms Collection of the Poznań University of Life Science (Department of Vegetable Crops) was used. The spawn (on wheat grain) for inoculation of substrates was prepared according to the method described by Stamets [55].

Wheat straw chaff (1–2 cm particle size) was used as the basic substrate, which was also the control substrate. The other combinations in the experiment were chaff with the following additions: wheat bran (WB) 20 g, thymus post-extraction waste (TPEW) 25 g and 50 g, pumpkin post-extraction waste (PPEW) 20 g per 100 g of basic substrate. The TPEW and PPEW were provided by Greenvit Company (Zambrów, Poland). The diversified addition of individual wastes was aimed at determining the possibility of their management in the cultivation of *P. citrinopileatus* in the absence of a clear crop licking. The differences in the percentage of waste addition resulted from the addition of various amounts of elements to the substrate, and therefore the possible, different effect on the growth of fruit bodies. These materials were obtained from thyme herb and crushed pumpkin seeds extracted with hot water (95–100 °C). All additives were dried to air-dry mass before use.

The substrates were wetted with distilled water to the 70% moisture content and were placed in 1 dm^3^ polypropylene bottles. The bottles with substrates were closed with caps with PP filters (class F-9, Filtropol, Oświęcim, Poland) and sterilised for one hour at a temperature of 121 °C. Afterward, the substrates were cooled to the temperature of 25 ± 1 °C. For inoculation, 10 g of spawn for each bottle was used.

The incubation was run at the temperature of 25 °C and 80–85% relative humidity until the substrate became completely covered with mycelium. Afterward, the caps were removed, and the bottles were placed in a cultivation facility. In this place, the air temperature was maintained at 21 ± 1 °C and relative humidity at 85–90%. The cultivation was additionally lighted with fluorescent light of 500 lx intensity for 12 h a day (12 h dark). The growth facility was aerated to maintain CO_2_ concentration below 1000 ppm. Fruit bodies were harvested successively as they matured. The whole fruit bodies were analysed.

### 2.2. Elemental Analysis

#### 2.2.1. Sample Preparation

Samples (both substrates and fruit bodies) were preliminarily dried in an electric oven (SLW 53 STD, Pol-Eko, Wodzisław Śląski, Poland) at the temperature 65 ± 2 °C for 120 h and ground (Cutting Mill PM 200 by Retsch GmbH, Haan, Germany) to determine dry weight (all results in this paper are expressed per dry weight of fruit bodies). Accurately weighed 0.500 ± 0.001 g of a dry mushroom sample was digested by concentrated (65%) nitric acid (Merck, Darmstadt, Germany) in sealed Teflon containers in the Mars 6 Xpress microwave sample preparation system. After digestion, samples were filtered through paper filters (Qualitative Filter Papers Whatman, Dassel, Germany, Grade 595: 4–7 µm) and diluted with water to the final volume of 15.0 mL. Each of the samples was analysed in triplicate using the whole sample preparation procedure.

#### 2.2.2. Instruments and Analytical Method Validation

Inductively coupled plasma optical emission spectrometry (Agilent 5110 ICP-OES, Agilent, Santa Clara, CA, USA) was used for multi-elemental analysis of the samples. The synchronous vertical dual view (SVDV) was used to provide simultaneously the axial and radial view of plasma. For multi-elemental determination the common spectrometric conditions were used: Radio Frequency (RF) power 1.2 kW, nebuliser gas flow 0.7 L min^−1^, auxiliary gas flow 1.0 L min^−1^, plasma gas flow 12.0 L min^−1^, viewing height for radial plasma observation 8 mm, CCD (Charge Coupled Device) detector temperature −40 °C, signal acquisition time 5 s for 3 replicates. For sample digestion, the Mars 6 microwave sample preparation system (CEM, Matthews, NC, USA) was applied.

For all the elements the detection limits were determined at 0.01–0.09 mg kg^−1^ dry weight (DW) (3-sigma criteria, Appendix A). The uncertainty for the total analytical procedure (including sample preparation) was below 20%. Traceability was checked using reference materials: CRM S-1—loess soil; CRM NCSDC (73349)—bush branches and leaves; CRM 2709—soil; CRM 405—estuarine sediments; CRM 667—estuarine sediments and the recovery (80–120%) was acceptable for most of the elements assayed. For non-certified elements, the recovery rate was determined in the standard addition method.

### 2.3. Analysis of Low-Molecular-Weight Organic Acids and Phenolic Compounds

The extraction method for low-molecular-weight organic acids (LMWOAs) and phenolic compounds in the studied powdered samples of *P. citrinopileatus* was presented in detail by Magdziak et al. [54] and Gąsecka et al. [45]. The obtained extracts were evaporated to dryness and stored frozen (−80 °C) until analysis. Prepared extracts before UPLC analysis were dissolved in 1 mL of deionized water for analysis of LMWOAs (Mili-Q, Millipore, Molsheim, France) and in 80% methanol for analysis of phenolic compounds, then centrifuged (at 3425× *g*) and filtered through filters (0.22 μm cellulose membrane) (MilliporeSigma Filtration, Burlington, MA, USA) immediately prior to chromatographic analysis. For the determination of LMWOAs and phenolic compounds 5 μL of liquor was injected onto UPLC (Acquity UPLC HSS T3 C18 column (150 mm × 2.1 mm, 1.8 μm particle size) (Waters, Dublin, Ireland) and photodiode array detector (PDA) eλ (Waters Corporation, Milford, MA, USA) and analysed as was described above for phenolic compounds [56]. A gradient program was applied with the mobile phase composed of A (water, containing 0.10% formic acid) and B (acetonitrile, containing 0.10% formic acid). External standards were used to quantify LMWOAs at λ = 280 nm wavelength, while phenolic acids were determined at λ = 280 nm and λ = 320 nm [56].

### 2.4. Determination of Total Phenolic Contents

Total phenolic contents were determined according to the standard procedure [57] with some modifications. A mixture of 1 mL of mushroom extract described above, 1 mL of Folin–Ciocalteu phenol reagent (diluted with H_2_O; 1:1, *v*/*v*) and 3 mL of 20% Na_2_CO_3_ was kept in the dark for 30 min at room temperature. Absorbance of the samples was then measured at 765 nm. The results were expressed as mg of gallic acid equivalent (GAE) per g of dry matter (DM).

### 2.5. Antioxidative Activity

The mushroom methanolic extracts from each type of cultivation at concentration between 0 and 20 mg mL^−1^ were mixed with 2.7 mL of 6 µmol L^−1^ 2,2-diphenyl-1-picrylhydrazyl (DPPH) radicals. After incubation in the dark at room temperature for 30 min absorbance at 517 nm was measured according to Dong et al. [58]. The scavenging activity of DPPH radicals was calculated according to the formula:Inhibition (%) = (A_blank_ − A_sample_)/A_blank_ × 100

A_blank_ is absorbance of the control (DPPH radical solution without extract), A_sample_ is absorbance of the methanolic extract of the mushroom.

### 2.6. The Statistical Analysis and Calculations

The obtained results are presented as mean values from triplicates and the standard deviation calculated by the STATISTICA 13.1 software (StatSoft, Tulsa, OK, USA). The differences between contents of nutritional major or trace elements, low-molecular-weight organic acids and phenolic compounds in different substrates were analysed with one-way analysis of variance (ANOVA) followed by post hoc Tukey’s test (*p* < 0.05). Pearson’s correlation (r) was estimated for total phenolic contents and DPPH scavenging ability with phenolic compounds.

To estimate the efficiency of element accumulation by *P. citrinopileatus*, the Bioconcentration Factor (BCF) was calculated as a ratio of element concentrations in the substrate to their contents in fruit bodies, both in dry matter.

## 3. Results

### 3.1. Morphology and Biomass of Mushroom Fruit Bodies

There were no differences in the shape and colour of *P. citrinopileatus* fruit bodies between the control and variants growing in enriched substrates (Figure 1). A significant difference was observed for biomass yield, where the addition of 25 or 50% TPEW or PPEW stimulated an increase in mushroom biomass (159, 163 and 150% of the control mushroom biomass, respectively). It is worth underlining that double supplementation with 50% TPEW non-significantly increased biomass yield in relation to the 25% addition of TPEW. The addition of WB caused a non-significant increase in *P. citrinopileatus* biomass (124% of the control).

### 3.2. Contents of Major Essential Elements

Generally, contents of major elements in individual experimental substrates were significantly different (Figure 2b), while contents of these elements, especially Ca, K and Mg, in fruit bodies were similar (Figure 2a). Obtained results show that the accumulation of major elements varied in mushrooms from substrates enriched with particular supplements. An example in this respect may be provided by K, where the highest content in straw was not reflected in the highest content of this element in fruit bodies. In substrates enriched with WB, PPEW and TPEW at 25 or 50%, significantly higher Ca contents were observed compared to the control (182, 203, 228 and 218% of the control, respectively).

Application of the supplements did not influence significantly stimulation/inhibition of K and Mg accumulation in fruit bodies, which is confirmed by the similar values of BCF (Table 1), is higher than 1, the same as for the control with an exception of mushrooms growing in the substrate enriched with 25% TPEW.

A clear stimulation of Na accumulation in mushrooms cultivated at substrates supplemented with PPEW and 25% TPEW was only observed (182 and 167% of the control, respectively). It is worth underlining that this increase was recorded only for mushrooms growing at 25% TPEW where BCF was 1.78, while for PPEW it was barely 0.72. For mushrooms growing in substrates enriched with WB, PPEW and 25% TPEW, a significantly higher content of S than in the control were also recorded (212, 256 and 244% of the control, respectively). Mushrooms from the substrate with a 50% TPEW addition were characterized by the same mean content of S as the control fruit bodies, while the BCF value was over 3-fold higher than for the control variant (1.16 and 0.37, respectively).

### 3.3. Contents of Trace Elements in Fruit Bodies

A varied accumulation of trace elements in mushrooms collected from the substrate enriched with particular supplements was also observed (Figure 3a, Figure 4a and Figure 5a). Their highest concentration in the used additions included to the substrate (Figure 3b, Figure 4b and Figure 5b) was not strictly correlated with their highest contents in fruit bodies, which was an effect of the presence of these elements both in straw and in the applied supplements. There were no significant differences between contents of Cu or Mo in the control and supplemented fruit bodies. Contents of Si, Sr and Te in the analysed mushrooms were almost the same with the exception of fruit bodies cultivated in the substrate enriched with WB (Figure 5a), where contents of these elements were significantly lower. A similar situation was observed for Ag and Sb with the exception of their significantly lower contents in mushrooms growing in the substrate enriched with 50% TPEW. The highest contents of Cr, Fe and Zn were observed in mushrooms collected from the substrate with the PPEW addition. Application of the supplements to the initial (control) substrate was the cause for the significantly lower accumulation of Bi, Cd, Ge, In, Ir, Mn, Nd, Ni, Os, Pb, Pr, Pt, Rb, Th, Ti, W and Zr in fruit bodies as compared with the control variant. In contrast, the significantly lower contents of Al, B (with the exception of mushrooms growing under 50% TPEW) and Ba were observed in the control variant in relation to mushrooms from the enriched substrates.

Values of BCF calculated for all the supplemented variants were higher than 1 for Cu, Te and Zn only. Generally, for most of the detectable trace elements, BCF > 1 was observed in the control variant, whereas for Ag, Al, B, Cr, Hg, Ni, Pb and Pt it was BCF < 1. The addition of especially 20% WB, but also 25 or 50% TPEW was the cause of the more effective uptake of Ag, Al, B and Cr compared to the control (Table 1).

### 3.4. Profile and Content of LMWOAs in Mushrooms

Depending on the post-extraction waste used, the samples of *P. citrinopileatus* fruit bodies were characterized by a diverse profile and content of determined LMWOAs (Table 2).

The lowest total LMWOAs contents were determined in the control 20% WB variants (0.05 and 0.18 µg g^−1^ DM, respectively). In the control variant, malonic and succinic acids were dominant, while for the 20% WB variant it was malic, succinic and quinic acids. The other acids were present at significantly lower levels or below the detection limits. In the variants where 20% PPEW, 25 and 50% TPEW were added, a significant increase was observed in the contents of studied acids (13.6, 24.4 and 112 µg g^−1^ DM, respectively). The 20% PPEW variant was characterised by the presence of five acids, with dominant citric and malonic acids, while oxalic, malic and fumaric acids were found at much lower levels. The substrate supplementation with TPEW caused significant changes in the studied LMWOAs contents. The highest levels of quinic, malic and citric acids and the sum of LMWOAs were observed. In the fruit bodies harvested from the 25% TPEW variant six acids (oxalic, quinic, malic, malonic, citric and succinic) were detected, with the dominance of citric, malonic and quinic acids, while in the 50% TPEW variant only three acids were found (quinic, malic and succinic acids).

### 3.5. Characteristics of the Phenolic Profile and Contents

Diversity in the phenolic profile was stated based on the results obtained from the UPLC analysis. Mushrooms growing in wheat straw chaff (control) were very poor in phenolic compounds in comparison to mushrooms cultivated in enriched substrates and only chlorogenic, gallic, syringic and vanillic acids were detected. The cultivation of mushrooms on enriched substrates resulted in quantitative and qualitative changes in phenolic profile. Mushrooms grown on WB20% showed the most diverse profile and contents of the phenolic compounds in comparison to the other supplemented variants (Table 3).

Fourteen phenolic compounds (caffeic, chlorogenic, *p*-coumaric, 2,5-dihydroxybenzoic (2,5-DHBA), ferulic, gallic, protocatechuic, salicylic, sinapic, syringic, *trans*-cinnamic and vanillic acids, catechin and rutin) were quantified in the fruit bodies grown on 20% WB, the variant with the highest total contents. The fruit bodies produced from substrates with 20% PPEW and 25% TPEW contained 10 phenolic compounds, but their composition varied. In mushrooms cultivated in the substrate with 50%, TPEW nine phenolic compounds were detected. Gallic and vanillic acids were recognised in all fruit bodies, while caffeic, *p*-coumaric and *trans*-cinnamic acids were quantified only in fruit bodies cultivated with 20% WB. The sums of the quantified phenolic compounds were highest for 20% WB, while they were lowest for the control variant. The sums of the phenolic compounds were similar for 20% PPEW and 50% TPEW.

The total phenolic contents (TPC) were very different ranging from 2.03 to 11.3 mg GAE g^−1^ DM. The lowest TPC was observed in mushrooms from the control variant, whereas supplementation stimulated an increase in TPC levels. The highest TPC was determined for cultivation with 20% WB. The total phenolic contents were comparable to the sum of the phenolic compounds, e.g., 20% WB > 50% TPEW ≥ 20% PPEW > 25% TPEW ≥ the control. The contents of all the acids were very highly correlated with total phenolic contents (0.988 ≥ r ≥ 0.970), except for 4-hydroxybenzoic acid (4-HBA). The scavenging ability of mushroom matter towards DPPH radicals varied for the composition of different substrates with an increase in the concentration of the extract (Figure 6).

The control variant shows the lowest scavenging effect of 43%. Generally, the addition of all waste materials to the substrates elevated the scavenging ability. The strongest effect of supplementation on the improvement of the scavenging ability (up to 80%) was observed for mushrooms cultivated with 20% PPEW. The ability to scavenge DPPH radicals was strongly correlated with TPC (*r* = 0.820) and with phenolic metabolites (0.989 ≥ *r* ≥ 0.969) except for 4-HBA, which confirmed the powerful role of phenolic compounds as antioxidants.

## 4. Discussion

According to Kalač [59], the number of commercially cultivated mushroom species is about 100 with 20 species cultivated on an industrial scale. For this reason, biofortification of not only *P. citrinopileatus*, but also many other species are important due to the improvement of their composition. Cultivation of *P. citrinopileatus* in substrates enriched with selected additions containing particular elements (e.g., Ca, Mg or Se) may be the cause of their increased transport during fructification [60,61]. However, it should be remembered that biofortification concerns only selected and essential elements such as Fe, Li, Se or Zn. For the cultivation of *P. citrinopileatus* in Poland, wheat or rye straw and deciduous sawdust enriched with cereal bran are mainly used to prepare the basic component in the culture medium [5,50]. Stimulation of mushroom growth on substrate enriched with organic wastes is related to the higher concentration of nitrogen as well as other necessary elements [62]. This was confirmed by results reported in a study by Singh and Singh [63], where *P. citrinopileatus* was cultivated separately or on various combinations of paddy straw with selected agriwastes (pea pod shell, radish, leaves, brassica straw or cauliflower leaves). Those authors observed that an optimal composition of the substrate (in this case 70% paddy straw and 30% of other agriwastes) stimulated *P. citrinopileatus* yield and bioefficiency. In our study, a higher or lower addition of TPEW (50 or 25%, respectively) was not the cause of significant changes in yield. Additionally, results described by Singh and Singh [63] that the preparation of substrate composed of an optimal mixture used in the experiment for WB, PPEW and/or TPEW could be a factor causing a significantly higher yield of this mushroom species. Owaid et al. [64] confirmed this point of view by indicating that the use of additives in the form of mixtures more effectively meets the nutritional requirements of mushrooms. This satisfaction of *P. citrinopileatus* nutritional needs to varying degrees with the wastes used in our studies can explain significant differences in the yield observed in Figure 1. The potential cause for differences in yields of *P. citrinopileatus* may have resulted from the varied ability of *P. citrinopileatus* to secrete hydrolytic and/or oxidative enzymes crucial in the use of lignocellulosic substrates [65].

The diverse ability of *P. citrinopileatus* to decompose substrates used in this experiment definitely resulted in different accumulation efficiency of major and trace elements. Both growth and transport of elements depended on growth conditions and nutrient type being the same for all the additives in our experiment. For this reason, both the initial concentration of elements in substrate additions and nutritional requirements are fundamental, which may be explained by significant differences in major elements in the supplemented substrates, especially K and Mg, but without significant differences in contents of these elements in mushrooms (Figure 2). In our experiment, concentrations of major elements in individual additions (WB, PPEW or TPEW) were adequate, because no negative symptoms were observed for their excess influencing the production of fruit bodies, described in the literature as the destruction of mycelium, reduction of yield or undesirable flavour of fruit bodies [66]. Additionally, it suggests that the nutritional requirements of *P. citrinopileatus* may be greater. The use of these waste mixtures could thus allow for introducing even larger amounts of major elements to the fruit bodies, without any health risk or possible reduction of biomass.

In the case of trace elements, a crucial factor explaining the observed differences between their concentrations in used wastes and contents in fruit bodies may be related to changes in mycelium pH [67]. In our experiment, the pH of mycelium for all waste additions ranged from 6.3 to 7.1, which is relatively optimal for the fructification process. A consequence of *P. citrinopileatus* colonisation was connected with decreasing pH, which could be the reason for the transport of bioavailable forms of trace elements from the wastes. Effective accumulation of trace elements from the substrate to fruit bodies depends mainly on substrate composition, with pH being a determinant of this transport. However, it should be remembered that the available elements are only a pool that the fruit bodies of *P. citrinopileatus* can use, while transport will probably be regulated by their nutritional needs. Differences in contents of essential trace elements (e.g., Cu or Zn) and trace elements with detrimental health effects (e.g., Ag or Hg) can confirm such an opinion. In the former group of elements, the efficiency of their accumulation changed up to a specific level, while for the latter group their transport was usually passive. It is worth underlining that similarly as in soil where, e.g., the concentration of bioavailable (soluble and exchangeable) forms of Cu is slight and their transport is effective within a proper range of pH [68], the same mechanism can modulate transport of detected trace elements from WB, PPEW and TPEW to fruit bodies. Moreover, during fructification mushrooms can limit the accumulation of especially toxic elements in the case of their excessive concentrations in the substrate as in the case of Ag or Pb and a 50% addition of TPEW (Figure 3 and Figure 4), or stimulate it, e.g., for Ba or Cd (Figure 3), when their concentration does not negatively affect fruit body development.

It should be remembered that the type of substrate and added waste determine not only the chemical composition of fruit bodies but also affect their functional and sensory properties, because these substrates may contain biologically active compounds. Studied LMWOAs are closely related to the synthesis of amino acids, phenols and esters, with the metabolism of aroma components [69]. In mushrooms acids additionally affect taste and aroma attributes [70], while they are also used as a factor extending their shelf life and suitability for consumption thanks to their antibacterial and antioxidant properties [71]. Pumpkin post-extraction wastes contain a high number of metabolites with biological (antidiabetic, antioxidant, anti-tumour and anti-inflammatory) properties [72,73], as well as sugars, minerals, amino acids (aspartic and glutamic acid and arginine) and carotenoids [74]. In turn, the presence of flavonoids, tannins, phenolic acids, LMWOAs, triterpene compounds, bitter substances, saponins, vitamins and minerals were detected in thyme extracts [75]. Similar properties were found for wheat and wheat bran being rich in compounds with a significant level of antioxidant activity, such as phenolic compounds [76]. Furthermore, wheat bran is an important by-product of the cereal industry that can be further used to improve technological efficiency [77]. The antioxidant activity of some durum wheat by-products is comparable to that of common fruit or fresh vegetables. However, to the best of our knowledge, no information is available on how added wastes affect the profile and contents of LMWOAs in *P. citrinopileatus*. The presented study shows that an addition of supplements caused differences in the profile of organic acids but especially stimulated their formation or production in fruit bodies. The addition of 20% WB had the lowest effect on the LMWOAs synthesis in *P. citrinopileatus*, for which a low increase in their content was noted compared to the control medium. However, the addition of 20% PPEW, 25 and 50% TPWE significantly contributed to the increase in LMWOAs contents in *P. citrinopileatus*. A confirmation of the influence of TPEW addition was provided by a study by Gąsecka et al. [56], where *P. eryngi* was cultivated on spent coffee grounds (SCG) and TPEW. The authors showed that the addition of SCG inhibited the formation of LMWOAs in *P. eryngi*, but the addition of TPEW significantly increased acid contents in fruit bodies, similar to the results obtained now for *P. citrinopileatus*. In the present study, apart from TPEW, also another post-extraction waste was used. Each of them caused an increase in the sum of the detected LMWOAs as compared to the control variant. This is highly valuable information indicating that the type of post-extraction waste added to the growing medium may have a significant impact on the decreasing or increasing contents of LMWOAs [56]. In the case of the LMWOAs increase in fruit bodies, it is a highly beneficial aspect because the determined acids constitute important molecules in mushrooms. In fruit bodies of *P. citrinopileatus* a significant increase in the contents of quinic, citric and malic acids was found following the supplementation with post-extraction wastes. The above-mentioned quinic acid is one of the antidiabetic ingredients, it shows antioxidant activity and is used to synthesise new pharmaceuticals [78,79]. Karamn et al. [79] in an in vitro screening study concerning the total antioxidant activity of various extracts from the mushroom *Coprinus comatus* found that quinic acid exhibits the most promising antioxidant potential. In view of the fact that quinic acid is commonly determined in mushroom fruit bodies, the potential to increase the content of this acid not only in *P. citrinopileatus* and/or *P. eryngii* [50], may definitely increase the importance of supplementation with the post-extraction waste to the cultivation media. In turn, citric acid protects mushrooms against browning thanks to its antibacterial and antioxidant properties [71] and it is also used to preserve fruit body colour. Gürgen and Yildiz [80] showed that pretreatment with citric acid in *P. citrinopileatus* helps to protect their yellow color when fruit bodies were dried. As for malic acid, it is often added to foods as an acidic additive to improve the organoleptic properties of foods [69]. The results showed in our work require extensive research in order to indicate what mechanism is associated with an increase in the contents of acids in fruit bodies. This information is very important in view of the fact that *P. citrinopileatus* is becoming increasingly popular thanks to its attractive colour and medicinal properties. Moreover, according to literature data, among the coloured mushrooms such as *P. citrinopileatus*, *P. ostreatus*, and *P. djamor*, the *P. citrinopileatus* strain exhibited the strongest antioxidant activity [80].

*Pleurotus* species are good sources of various phenolic compounds [44,45,81,82], the content and composition of which may be modified by the addition of different supplements to the growing substrate [83,84,85,86]. Phenolic compounds were recognized as major antioxidant components of *P. citrinopileatus* [22,82]. It is difficult to state the content of total phenolics because the range was very wide and dependent on temperature, the solvent used in the extraction and the method of extraction [24,81,87]. In a study by Gogoi et al. [81], it was between 361.78 to 678.06 mg GAE 100 g^−1^ DM. A much lower total phenolic content (64 µg catechol DW) was determined by Khatun et al. [24]. In turn, Lee et al. [22] reported contents between 8.62 and 12.38 mg g^−1^ DM, while Rodrigues [25] recorded 1140 μg cathecol equiv. g^−1^ dry mushroom. Additionally, it was documented that the primordial phases of *P. citrinopileatus* exhibited a higher TP at 13.8 mg GAE g^−1^ than in matured fruit bodies [82]. The results obtained in the present study for the control variant are lower than many of the above-mentioned reports.

The chromatographic profile in the control variant showed only chlorogenic, gallic, syringic and vanillic acids. The poor composition was also reported by Gogoi et al. [81], who detected only gallic acid, caffeic acid, rutin and quercetin, while Nattoh et al. [82] quantified a large number of phenolic compounds such as kaempferol, hesperetin, biochanin A, naringin, dinoprostone, resveratrol, loganate, formononetin, roseoside as well as ferulic, 4-phenylbutyric, coumaric, homogentisic, gallic, protocatechuic and caffeic acids, depending on the phenological phase of *P. citrinopileatus*.

The changes in phenolic contents and composition as well as in the scavenging ability of fruit bodies were strongly exhibited after the enrichment of the substrate with different waste materials. All the used supplements elevated total phenolic contents, which is a positive effect because of their antioxidative activity. Especially the increase up to 11.3 mg GAE g^−1^ DM was seen in the substrate with the 20% WB addition. A similar acceleration was observed in the contents and composition of phenolic compounds. A more diverse composition of phenolics as compared with literature data, indicating up to 14 compounds, was found in fruit bodies grown on the supplemented substrates. Some studies researched the effect of substrate supplementation with waste materials on bioactive compounds in mushrooms, producing inconsistent results [83,84,85,86]. Coffee by-products picked up elevated TPC from hot water and enzymatic extracts, but not from the ethanolic extract in *P. citrinopileatus*, while opposite results were observed for *P. salmoneo-stramineus* in the hot water extract [86]. No changes in TPC and antioxidant activity were noted in *Pleurotus* species cultivated on the substrate supplemented with waste obtained from the two-phase olive oil production [83]. An opposite opinion stating that the chemical composition and substrate nature significantly affect levels of bioactive compounds in fruit bodies was confirmed in other studies [50,84,88,89]. Singh et al. [84] showed that different agri-residues strongly affect the modification of phenolic contents and improvement of antioxidant activity in *P. ostreatus.* Phenolic compounds in *Pleurotus* species were sensitive to the addition of different residues (grape marc, olive mill byproducts and wheat straw) into the substrate, which strongly increased contents of phenolic acids, resveratrol, total phenolics and improved antioxidant activity (2- to 8-fold increase) [89]. A significant effect on the increase in phenolic contents and antioxidant activity was found also for cotton waste added to a substrate used in the cultivation of *Pleurotus* species [85,90]. A strong effect for the supplementation of the growing substrate was confirmed also for other mushroom species. Some isolates of *Hericium erinaceus* were strongly affected by the addition of olive press cake as regards total phenolic contents and antioxidant activity, while it was not the case for others [46]. A powerful effect of olive pruning residues in the substrate of *H. erinaceus* on total phenolics was confirmed by Koutrotsios et al. [89]. Ruiz-Rodriguez et al. [83] stated that no compounds from the residue were detected in fruit bodies. A clearly visible positive effect of some substrates on the composition and content of some bioactive compounds was found in these results. However, it may not be excluded that extraction parameters (solvent, temperature, etc.) may also indicate other results than those cited in the literature. The increase in contents of phenolic compounds and total phenolics is also important, because of the significant correlation with the scavenging ability (0.734 < r < 0.989), while studies proved that relationships between supplementation, phenolic contents and antioxidant properties. Thus, the mechanisms should be studied in further research to optimise the cultivation, especially because antioxidant compounds have a beneficial effect on human health and the quality of food products because they prevent oxidative spoilage. Although the fruiting bodies obtained from substrate supplemented with WB 20% contained a much greater amount of phenolic compounds, their ability to scavenge radicals was not the greatest. This could be due to the content of other bioactive compounds in mushrooms harvested in the other enriched substrates, which also have antioxidant activity.

## 5. Conclusions

In summary, the use of organic post-extract waste materials can significantly stimulate the production of LMWOAs in fruit bodies of *P. citrinopileatus*, thereby increasing their biological value.

Mycelia of *P. citrinopileatus* cultivated under different conditions showed significant differences in growth and metabolic profiles of phenolic compounds and LMWOAs, which may be used in the next step in special cultivation, e.g., for medicinal uses or for human consumption. Moreover, supplementation of the growing substrate with wheat bran, thymus post-extraction waste and pumpkin post-extraction waste is a good method to increase contents of LMWOAs and phenolic compounds, being important natural antioxidants, and thus to improve their antioxidant capacity.

## Figures and Tables

**Figure 1 jof-07-00894-f001:**
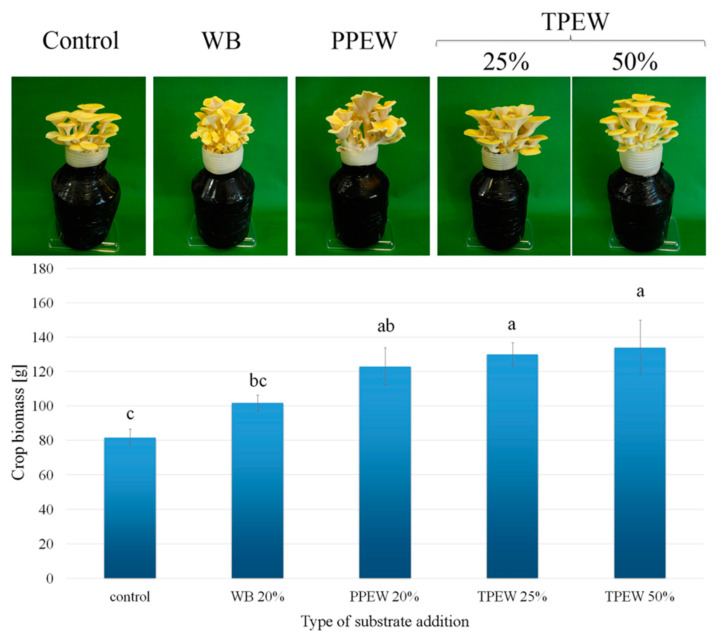
Morphology and biomass crop (g) of *Pleurotus citrinopileatus* growing under particular additions to substrate. Mean values (*n* = 3); identical superscripts (^a,b,c^) denote no significant (*p* < 0.05) difference between mean values in columns according to Tukey’s HSD test (ANOVA) for mushrooms growing at particular substrates. WB—wheat bran, PPEW—pumpkin post-extraction waste, TPEW—thymus post-extraction waste.

**Figure 2 jof-07-00894-f002:**
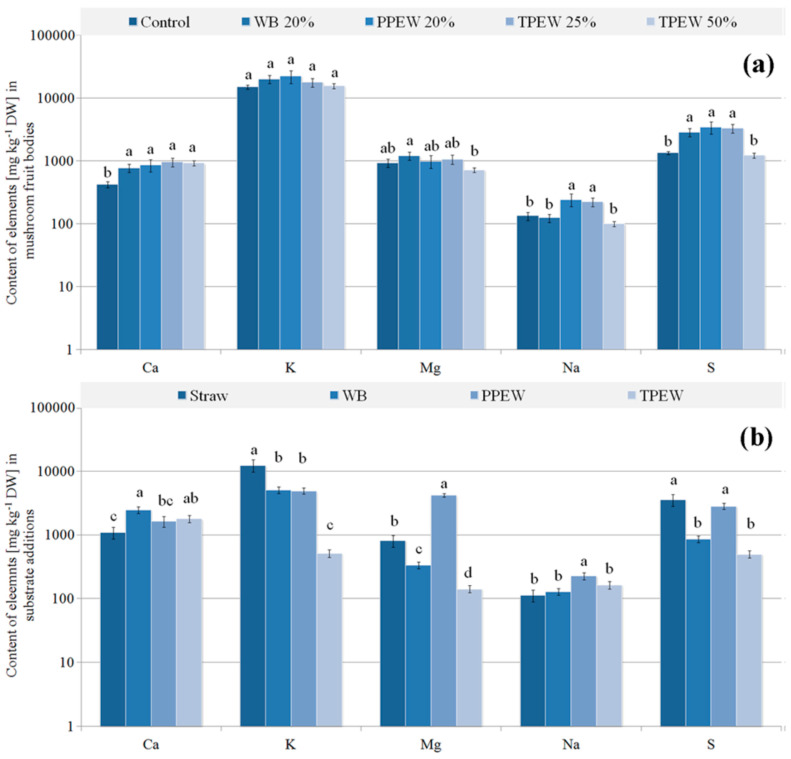
Content of nutritional elements (mg kg^−1^ DM) in *Pleurotus citrinopileatus* fruit bodies (**a**) and particular additions included to substrate (**b**). Mean values (*n* = 3); identical superscripts (^a–d^) denote no significant (*p* < 0.05) difference between mean values in columns according to Tukey’s HSD test (ANOVA) for mushrooms growing at particular substrates. WB—wheat bran, PPEW—pumpkin post-extraction waste, TPEW—thymus post-extraction waste.

**Figure 3 jof-07-00894-f003:**
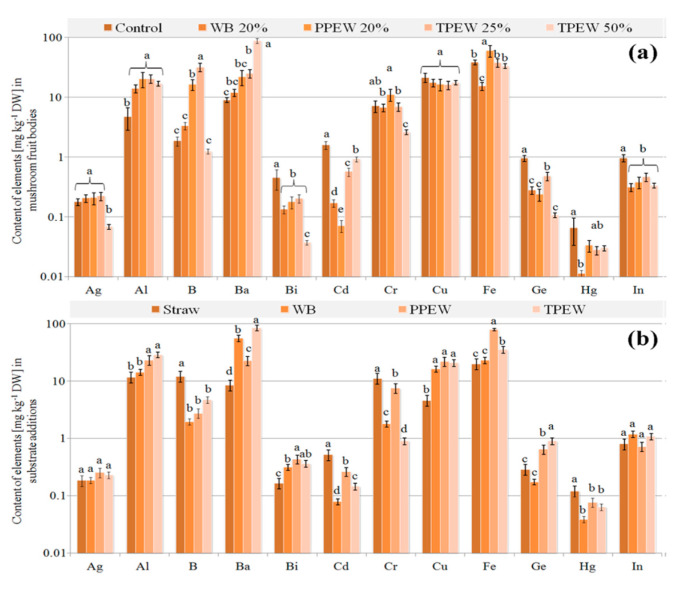
Content of trace elements (mg kg^−1^ DM) in *Pleurotus citrinopileatus* fruit bodies (**a**) and particular additions included to substrate (**b**). Mean values (*n* = 3); identical superscripts (^a–d^) denote no significant (*p* < 0.05) difference between mean values in columns according to Tukey’s HSD test (ANOVA) for mushrooms growing at particular substrates. WB—wheat bran, PPEW—pumpkin post-extraction waste, TPEW—thymus post-extraction waste.

**Figure 4 jof-07-00894-f004:**
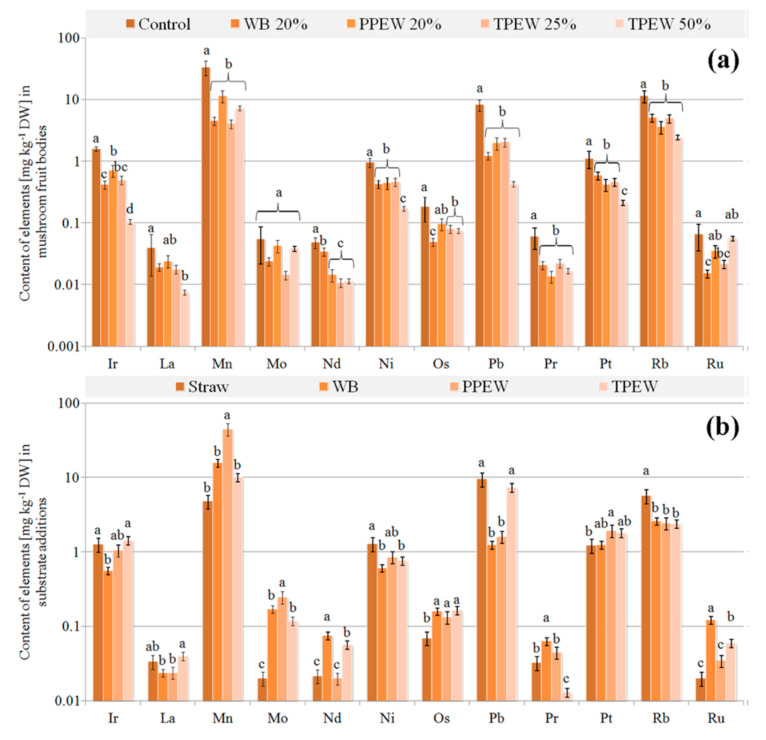
Content of trace elements (mg kg^−1^ DM) in *Pleurotus citrinopileatus* fruit bodies (**a**) and particular additions included to substrate (**b**). Mean values (*n* = 3); identical superscripts (^a–d^) denote no significant (*p* < 0.05) difference between mean values in columns according to Tukey’s HSD test (ANOVA) for mushrooms growing at particular substrates. WB—wheat bran, PPEW—pumpkin post-extraction waste, TPEW—thymus post-extraction waste.

**Figure 5 jof-07-00894-f005:**
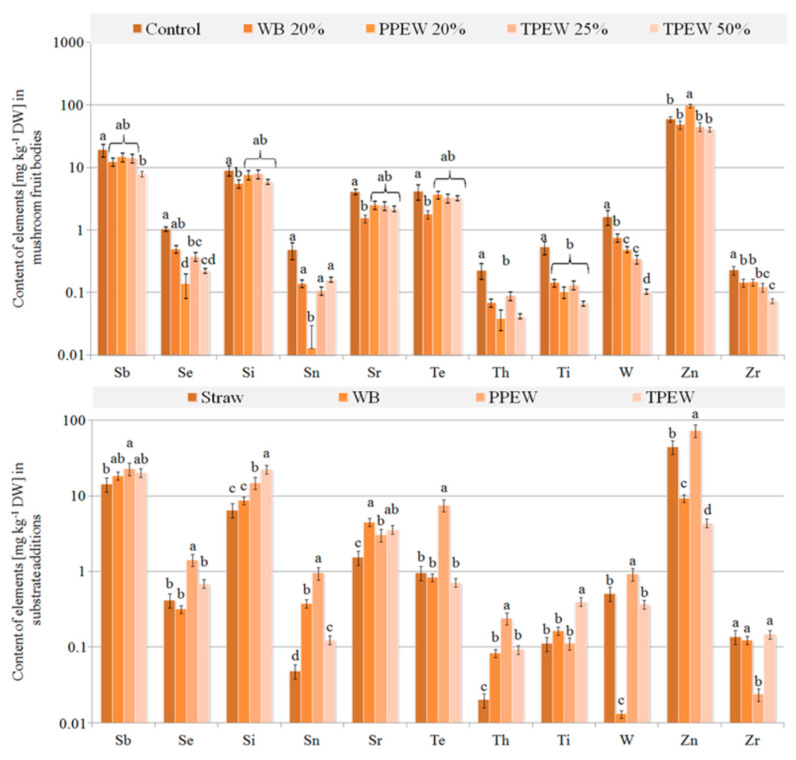
Content of trace elements (mg kg^−1^ DM) in *Pleurotus citrinopileatus* fruit bodies (**a**) and particular additions included to substrate (**b**). Mean values (*n* = 3); identical superscripts (^a–d^) denote no significant (*p* < 0.05) difference between mean values in columns according to Tukey’s HSD test (ANOVA) for mushrooms growing at particular substrates. WB—wheat bran, PPEW—pumpkin post-extraction waste, TPEW—thymus post-extraction waste.

**Figure 6 jof-07-00894-f006:**
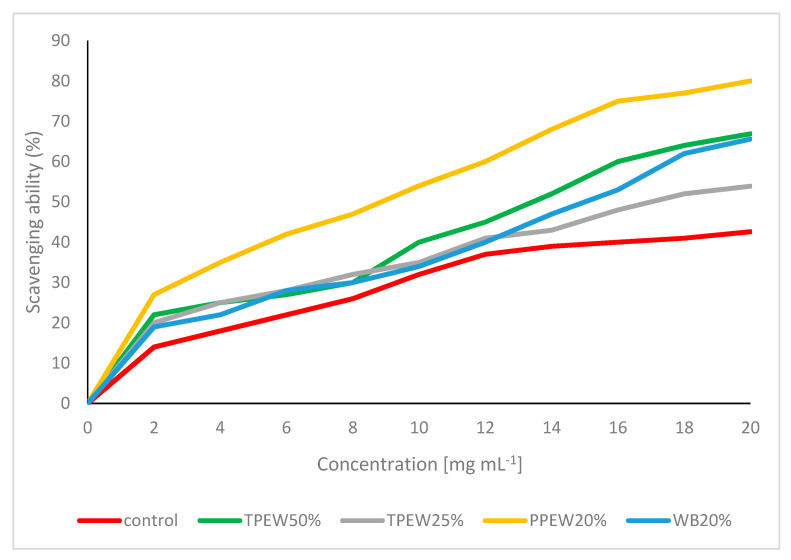
Scavenging ability on DPPH radicals of methanolic extract of *Pleurotus citrinopileatus*.

**Table 1 jof-07-00894-t001:** Bioconcentration factor (BCF) values calculated for *Pleurotus citrinopileatus* growing at particular substrates.

Element	Control	WB 20%	PPEW 20%	TPEW 25%	TPEW 50%
Ca	0.39	0.56	0.64	0.71	0.76
K	**1.20**	**1.83**	**2.42**	**2.03**	**1.90**
Mg	**1.13**	**1.66**	**1.49**	0.66	**1.63**
Na	**1.18**	**1.07**	0.72	**1.78**	**1.77**
S	0.37	0.94	0.60	**1.00**	**1.16**
Ag	0.95	**1.11**	0.33	**1.04**	**1.13**
Al	0.40	**1.14**	0.83	**1.44**	**1.27**
B	0.15	0.33	0.15	**1.60**	**3.11**
Ba	**1.06**	0.67	**1.88**	**1.92**	0.91
Bi	**2.69**	0.67	0.14	0.80	0.94
Cd	**3.05**	0.39	**2.75**	0.15	**1.33**
Ce	**4.43**	**2.65**	**1.30**	0.77	0.72
Cr	0.65	0.73	0.44	**1.07**	0.81
Cu	**4.66**	**2.54**	**1.39**	**2.04**	**1.86**
Er	**5.3**1	**1.14**	**2.18**	0.76	**2.06**
Fe	**1.92**	0.75	**1.21**	**1.88**	**1.5**9
Ge	**3.29**	**1.06**	0.18	0.65	**1.09**
Hf	**2.73**	0.16	0.30	0.75	0.38
Hg	0.53	0.11	0.32	0.30	0.26
In	**1.19**	0.36	0.35	0.48	0.53
Ir	**1.25**	0.37	0.08	0.58	0.38
La	**1.17**	0.61	0.20	0.76	0.50
Mn	**6.96**	0.65	0.98	0.90	0.67
Mo	**2.74**	0.48	0.56	0.66	0.32
Nd	**2.24**	**1.06**	0.29	0.67	0.35
Ni	0.74	0.37	0.17	0.37	0.40
Os	**2.64**	0.56	0.64	**1.17**	0.86
Pb	0.86	0.16	0.05	0.25	0.23
Pr	**1.86**	0.53	0.73	0.39	0.80
Pt	0.90	0.48	0.14	0.31	0.34
Rb	**2.02**	**1.00**	0.60	0.71	**1.02**
Ru	**3.26**	0.38	**1.43**	**1.52**	0.72
Sb	**1.3**3	0.82	0.46	0.91	0.89
Se	**2.46**	**1.24**	0.40	0.22	0.77
Si	**1.38**	0.79	0.41	0.93	0.75
Sn	**9.97**	**1.22**	**1.86**	0.06	**1.59**
Sr	**2.67**	0.72	0.85	**1.37**	**1.19**
Te	**4.32**	**1.91**	**3.88**	**1.59**	**3.59**
Th	**11.3**	**2.09**	0.73	0.60	**2.31**
Ti	**4.79**	**1.17**	0.26	0.91	0.71
W	**3.18**	**1.82**	0.23	0.82	0.72
Zn	**1.33**	**1.28**	**1.67**	**1.93**	**1.29**
Zr	**1.65**	**1.07**	0.52	**1.27**	0.87

Bold shows an effective accumulation of elements. A normal is used for exclusion strategy.

**Table 2 jof-07-00894-t002:** Content of aliphatic low-molecular organic acids (LMWOAs) (µg g^−1^ DM) in fruit bodies of *Pleurotus citrinopileatus* growing at substrates enriched with different supplements.

Acid	Control	WB 20%	PPEW 20%	TPEW 25%	TPEW 50%
Acetic	nd	0.129 ^a^ ± 0.0632	nd	nd	nd
Citric	0.941 ^d^ ± 0.0171	nd	689 ^c^ ± 33.7	1055 ^b^ ± 19.4	5536 ^a^ ± 239
Fumaric	nd	nd	7.49 ^a^ ± 0.366	nd	nd
Maleic	nd	0.587 ^a^ ± 0.0287	nd	nd	nd
Malic	nd	10.5 ^c^ ± 0.211	134 ^b^ ± 6.50	49.9 ^c^ ± 6.93	3262 ^a^ ± 138
Malonic	1.92 ^c^ ± 0.350	nd	471 ^b^ ± 23.0	663 ^a^ ± 12.2	nd
Oxalic	nd	0.148 ^c^ ± 0.0724	53.9 ^a^ ± 2.61	41.8 ^b^ ± 3.77	nd
Quinic	0.609 ^c^ ± 0.0112	2.36 ^c^ ± 0.115	nd	581 ^b^ ± 10.7	2431 ^a^ ± 105
Succinic	1.71 ^b^ ± 0.000311	4.11 ^b^ ± 0.201	nd	49.7 ^a^ ± 5.9	nd
Sum	5.19 ^d^ ± 0.948	17.8 ^d^ ± 1.871	1355 ^c^ ± 66.2	2440 ^b^ ± 145	11229 ^a^ ± 485

Mean values (*n* = 3); identical superscripts (^a–d^) denote no significant (*p* < 0.05) difference between mean values in rows according to Tukey’s HSD test (ANOVA) for mushrooms growing at particular substrates. WB—wheat bran, PPEW—pumpkin post-extraction waste, TPEW—thymus post-extraction waste, nd—not detected.

**Table 3 jof-07-00894-t003:** Content of phenolic compounds (µg g^−1^ DM) in fruit bodies of *Pleurotus citrinopileatus* growing at substrates enriched with different supplements.

Compound	Control	WB 20%	PPEW 20%	TPEW 25%	TPEW 50%
Caffeic	nd	151 ± 11.9	nd	nd	nd
Catechin	nd	407 ^a^ ± 14.1	25.9 ^b^ ± 1.83	20.2 ^b^ ± 0.919	2.21 ^c^ ± 0.171
Chlorogenic acid	2.73 ^b^ ± 0.143	853.4 ^a^ ± 0.00775	3.23 ^b^ ± 0.0841	nd	nd
*p*-Coumaric acid	nd	251 ± 21.1	nd	nd	nd
2,5-DHBA	nd	2162 ^a^ ± 112.3	33.7 ^b^ ± 0.757	23.3 ^b^ ± 02.67	55.8 ^b^ ± 4.76
Ferulic acid	nd	152 ^a^ ± 04.50	nd	2.0 ^b^ ± 0.151	2.10 ^b^ ± 0.553
Gallic acid	13.2 ^b^ ± 0.989	2190 ^a^ ± 149	60.7 ^b^ ± 01.32	36.0 ^b^ ± 1.38	148 ^b^ ± 21.6
4-HBA	nd	nd	143^a^ ± 14.5	66.1 ^b^ ± 1.87	nd
Protocatechuic acid	nd	622 ^a^ ± 16.1	22.3 ^b^ ± 0.481	23.6 ^b^ ± 0.745	24.3 ^b^ ± 3.61
Rutin	nd	497 ^a^ ± 25.4	nd	3.07 ^b^ ± 0.121	2.98 ^b^ ± 0.165
Salicylic acid	nd	858 ^a^ ± 38.5	1.63 ^b^ ± 0.135	7.69 ^b^ ± 0.455	6.90 ^b^ ± 0.177
Sinapic acid	nd	331 ^a^ ± 18.1	1.92 ^b^ ± 1.28	nd	nd
Syringic acid	16.5 ^b^ ± 1.26	298 ^a^ ± 12.9	15.5 ^bc^ ± 1.32	9.12 ^bc^ ± 0.309	nd
*trans*-Cinnamic acid	nd	23.3 ± 2.91	nd	nd	nd
Vanillic acid	4.01 ^b^ ± 0.209	1690 ^a^ ± 131	57.1 ^b^ ± 2.42	77.9 ^b^ ± 4.05	145 ^b^ ± 1.5
Total phenolic content [mg GAE g^−1^ DM]	2.03 ^c^ ± 0.137	11.3 ^a^ ± 1.03	3.22 ^bc^ ± 0.300	2.22b ^c^ ± 0.203	3.42 ^b^ ± 0.321

Mean values (*n* = 3); identical superscripts (^a–c^) denote no significant (*p* < 0.05) difference between mean values in rows according to Tukey’s HSD test (ANOVA) for mushrooms growing at particular substrates 2,5-DHBA—2,5-dihydroxybenzoic acid, 4-HBA—4-hydroxybenzoic acid. WB—wheat bran, PPEW—pumpkin post-extraction waste, TPEW—thymus post-extraction waste, nd—not detected.

## Data Availability

Not applicable.

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
