# Peer review of "A Possibility to Use Selected Crop Post-Extraction Wastes to Improve the Composition of Cultivated Mushroom Pleurotus citrinopileatus"

_jof, 2021, doi:10.3390/jof7110894_

Round 1
Reviewer 1 Report
An interesting paper concerning the cultivated mushrooms. I recommend the publication.
Author Response
Dear Reviewer#1,
we cordially thanks to Reviewer#1 for review and acceptance our paper. We are very grateful for such a positive opinion on our manuscript.
Yours faithfully,
The entire team of authors
Reviewer 2 Report
The paper entitled '' A possibility to use selected crop post-extraction wastes to prove the composition of cultivated mushroom Pleurotus citrinopileatus'' examine the potentials of the use of agrowastes as mushroom cultivation substrates. The present work is very well-written and adequately supported by experimental and scientific facts. The authors should address few issues cited below:
1) Why the authors used different percentages of wastes (20% vs 25 and 50%) for pumpkin and thyme?
2) Lines 251-259: Is the pH value the only determinant factor for the differences in the mineral content or the variances may have other possible explanations?
3) Line 314: According to literature, thyme contains various phenolic compounds. Do they authors have any assumptions regarding the presence of less phenolics in the fruit bodies cultivated in such a phenolic-rich substrate?
Author Response
Dear Reviewer#2,
we greatly appreciated for all your comment and suggestion. We took all them into account, and we marked all changes in green. Thank you very much for your time, review of our work, and all suggestion, which made our manuscript better. All changes in the text are marked in green.
We attached the responses and comments to the Reviewer's suggestions in a separate file.
Yours faithfully,
All Authors
Thank you very much for your time, review of our work, and all suggestion, which made our manuscript better. Below we present our answers.
- Why the authors used different percentages of wastes (20% vs 25 and 50%) for pumpkin and thyme?_
Author response:The different amount of thyme and coffee waste added was due to several reasons. The materials used in experiment contain a certain amount of organic substances, including especially organic nitrogen. We wanted to find out to what extent these wastes affect the growth and yielding of oyster mushroom as an addition to the substrate (positive or negative). It is also known that both thyme and coffee contain a number of substances (for example essential oils or caffeine) that could have a negative effect on the oyster mushroom. Moreover, it is also known that some essential oils can inhibit the growth of mycelium, e.g. Penicillium or Trichoderma. The diversification of the addition of these wastes to the substrate allowed for a preliminary assessment of whether it is possible to utilize them with Pleurotus citrinopileatus, taking into account the effective yielding. We also added some information in “Experimental material and experimental design” section in line 102-107.
- Lines 251-259: Is the pH value the only determinant factor for the differences in the mineral content or the variances may have other possible explanations?
- Author response: Environmental factors in case of field experiments, the same as the influence of the matrix in our experiment is not limited to pH. The effects of the content of organic matter, the value of electrolytic conductivity as a measure of salinity or even particle size are widely described. However, these measurements have not been made and writing about them would be a speculation rather than a reliable assessment of the state of affairs. Your question is very valuable. We have already overtaken this in subsequent studies carried out on load, where a full analysis of the substrates used is performed. Thank you.
- Line 314: According to literature, thyme contains various phenolic compounds. Do they authors have any assumptions regarding the presence of less phenolics in the fruit bodies cultivated in such a phenolic-rich substrate?
Author response: The table 2 and 3 unfortunately contains a mistake in the unit. The proper unit is µg g-1 DM. We would like sincerely apologize for this errors, which could suggested that the bioactive compounds are in very low content in fruiting bodies. The mistake was improved. The obtained results shows that control mushrooms had lower content of phenolic compounds and moreover the profile was less diverse that in mushrooms cultivated in enriched substrates. The use of the enriched substrates caused the quality and quantity changes in phenolic profile.

Reviewer 3 Report
The paper aimed to analysis the possibility to use selected crop post-extraction waste to improve the composition of cultivated mushroom Pleurotus citrinopileatus. The study showed that the post-extraction wastes are a crucial factor determining the accumulation of minerals, the content/profile of low-molecular weight organic acids (LMWOAs) and phenolic compounds in fruit bodies, thereby increasing their nutritional value. This study is significant for both of the edible mushroom industry and agriculture. However, there were several serious questions in this study.
- The Introduction part can be appropriately deleted.
- The part of Materials and methods is not well-structured, and the logic is not strong. It should be for example 2.1 Sample preparation 2.2 Analytical methods 2.3 Antioxidative activity 2.4 Statistical analysis.
- line107-110: the amount of wheat bran (WB), thymus post-extraction waste (TPEW), and pumpkin post-extraction waste (PPEW) that the authors added to the base substrate should be clarify and reasonable. According the results, the amounts of WB, TPEW and PPEW were respectively 20%, 25%, 50% and 20%, rather than 20g, 25g and 50g, 20g per 100g of basic substrate.
Why was the amounts of WB, TPEW and PPEW being respectively of 20%, 25%, 50% and 20%?
- line299: as shown in 3.5, fourteen phenolic compounds were quantified in the fruit bodies grown on 20% WB, and its total phenolic content up to 11.3mg GAE/g DM. But why is it scavenging ability on DPPH radical is inferior to that of mushrooms cultured by 20% PPEW and 50% TPEW.
- It is recommended to remove the horizontal line in the broken line chart of Figure 6.
Author Response
Dear Reviewer,
Thank you very much for such a thorough analysis of our work, for your time and all suggestions. We tried to answer the best we could. If there were any other comments, we are ready for changes and cooperation. All corrections in the text are marked in green.
I attach the answers to all suggestions in an additional file. Once again, we would like to thank the Reviewer for suggestion which made our work better.
Yours faithfully,
All Authors.
Comments and Suggestions for Authors
The paper aimed to analysis the possibility to use selected crop post-extraction waste to improve the composition of cultivated mushroom Pleurotus citrinopileatus. The study showed that the post-extraction wastes are a crucial factor determining the accumulation of minerals, the content/profile of low-molecular weight organic acids (LMWOAs) and phenolic compounds in fruit bodies, thereby increasing their nutritional value. This study is significant for both of the edible mushroom industry and agriculture. However, there were several serious questions in this study.
- The Introduction part can be appropriately deleted.
Author response: Introduction has been corrected. We've removed those snippets that we don't think actually add much to the work. We hope that the current form will be satisfactory. If, however, there are still comments / suggestions, we are ready for changes.
- The part of Materials and methods is not well-structured, and the logic is not strong. It should be for example 2.1 Sample preparation 2.2 Analytical methods 2.3 Antioxidative activity 2.4 Statistical analysis.
Author response: The part of Materials and methods was not well-structured, and the logic was not strong - thank You very much for your suggestion.
We have tried to organize this part of the work as well as possible. We have created the following sections: 2.1. Experimental material and experimental design, 2.2. Elemental analysis 2.2.1 Sample preparation, 2.2.2 Instruments and analytical method validation, 2.3. Analysis of low-molecular weight organic acids and phenolic compounds, 2.4. Determination of total phenolic contents, 2.6 Antioxidative activity, 2.7 The statistical analysis and calculations
We know that we did not follow the example of how to arrange this point according to the reviewer's guidelines, but we wanted to avoid duplicate headlines.
However, if our point of view and the proposed changes are not accepted by the reviewer, we are ready for further changes.
- line107-110: the amount of wheat bran (WB), thymus post-extraction waste (TPEW), and pumpkin post-extraction waste (PPEW) that the authors added to the base substrate should be clarify and reasonable. According the results, the amounts of WB, TPEW and PPEW were respectively 20%, 25%, 50% and 20%, rather than 20g, 25g and 50g, 20g per 100g of basic substrate. Why was the amounts of WB, TPEW and PPEW being respectively of 20%, 25%, 50% and 20%?
Author response: The different amount of thyme and coffee waste added was due to several reasons. The materials used in experiment contain a certain amount of organic substances, including especially organic nitrogen. We wanted to find out to what extent these wastes affect the growth and yielding of oyster mushroom as an addition to the substrate (positive or negative). It is also known that both thyme and coffee contain a number of substances (for example essential oils or caffeine) that could have a negative effect on the oyster mushroom. Moreover, it is also known that some essential oils can inhibit the growth of mycelium, e.g. Penicillium or Trichoderma. The diversification of the addition of these wastes to the substrate allowed for a preliminary assessment of whether it is possible to utilize them with Pleurotus citrinopileatus, taking into account the effective yielding. We also added some information in “Experimental material and experimental design” section in line 102-107.
- line299: as shown in 3.5, fourteen phenolic compounds were quantified in the fruit bodies grown on 20% WB, and its total phenolic content up to 11.3mg GAE/g DM. But why is it scavenging ability on DPPH radical is inferior to that of mushrooms cultured by 20% PPEW and 50% TPEW.
Author response: The scavenging ability is correlated not only with phenolic compounds , but also with other bioactive compounds. In the manuscript we added the sentence explaining the this fact:
Although the fruiting bodies obtained from substrate supplemented with WB 20% contained a much greater amount of phenolic compounds, their ability to scavenge radicals was not the greatest. This could be due to the content of other bioactive compounds in mushrooms harvested in the other enriched substrates, which also have antioxidant activity.
- It is recommended to remove the horizontal line in the broken line chart of Figure 6.
Author response: The lines were removed.

Round 2
Reviewer 3 Report
OK